# Effect of Brewing Methods on Acrylamide Content and Antioxidant Activity: Studying Eight Different Filter Coffee Preparations

**DOI:** 10.3390/antiox12101888

**Published:** 2023-10-21

**Authors:** Agnese Santanatoglia, Simone Angeloni, Davide Bartolucci, Lauro Fioretti, Gianni Sagratini, Sauro Vittori, Giovanni Caprioli

**Affiliations:** 1Chemistry Interdisciplinary Project (ChIP), School of Pharmacy, University of Camerino, Via Madonna delle Carceri 9/B, 62032 Camerino, Italy; agnese.santanatoglia@unicam.it (A.S.); simone.angeloni@unicam.it (S.A.); gianni.sagratini@unicam.it (G.S.); sauro.vittori@unicam.it (S.V.); 2Research and Innovation Coffee Hub, Via Emilio Betti 1, 62020 Belforte del Chienti, Italy; lauro.fioretti@simonelligroup.it; 3Chemistry Interdisciplinary Project (ChIP), School of Science and Technology, University of Camerino, Via Madonna delle Carceri 9/B, 62032 Camerino, Italy; davide.bartolucci@studenti.unicam.it

**Keywords:** acrylamide, antioxidant activity, brewing methods, chlorogenic acids, coffee, HPLC-MS/MS, HPLC-DAD

## Abstract

The aim of this study was to investigate the parameters affecting the extraction of positive molecules such as chlorogenic acids and antioxidants, as well as potentially carcinogenic substances such as acrylamide, in different coffee brewing methods. Three coffee varieties, each assigned a different roasting degree, were used to prepare coffee brews following eight different preparation methods. Acrylamide was quantified using the HPLC-MS/MS instrument, while chlorogenic acids and caffeine were quantified using the HPLC-DAD system. Three spectrophotometric analyses were also performed (DPPH, TFC and TPC) to evaluate antioxidant activity. The results showed that the main brewing parameters, which have the greatest influence on the final content of these compounds, were the volume of water used, more specifically the brewing ratio (coffee to water ratio), the extraction time and the particle size of the coffee powder. In addition, the variables that have the greatest impact on the discrimination of the preparation methods studied are total chlorogenic acid content, TFC, TPC, caffeine and the DPPH assay. For this reason, the recipe and infusion parameters used for each of the extraction systems are the key factors that determine the extraction of coffee components and, consequently, the quality of the cup.

## 1. Introduction

Coffee, a beverage widely consumed around the world, is obtained from the seeds of some plants of the genus *Coffea*. The popularity of this beverage is likely due to its distinctive organoleptic and sensory properties, some of its bioactive constituents and some of its beneficial effects on human health. During the heat treatment that occurs when green coffee beans are roasted, numerous transformations take place that belong to the Maillard reaction network or non-enzymatic browning.

### 1.1. Acrylamide in Coffee

Reducing sugars and amino acids are involved in these processes, which occur at high temperatures (at least 120 °C) under conditions of low water activity [1]. The Maillard reaction plays an important role in improving the appearance and taste of foods and it is related to aroma, flavor and color [2]; at the same time, potential toxic substances can be formed during this reaction. One of these is acrylamide (AA), a substance classified as probably carcinogenic to humans according to IARC (International Agency for Research on Cancer) and belonging to group 2A. The EFSA (European Food Safety Authority) has determined, based on data from animal studies, that the presence of AA in foods may increase the risk of cancer for consumers of all ages, and therefore it is recommended that the AA intake should be limited. After ingestion, acrylamide is converted to glycidamide, and this appears to be the most likely cause of the establishment of gene mutations that promote the development of cancer [3]. The main pathway leading to the formation of AA in coffee and food involves the interaction of the amino group of free asparagine (not bound in amino acid chains) condensing with the carbonyl group of reducing sugars (fructose, glucose, galactose, etc.) [4]. AA is a polar substance, and it is easily extracted with hot water during the preparation of coffee. These beverages are obtained by infusion or percolation of roasted and ground coffee with hot water, therefore, depending on the recipe or extraction method used in the preparation of the coffee, the content of extracted acrylamide from coffee powder can vary [5]. In detail, a factor that affects the extraction of AA and more broadly the chemical composition of the brew, is the volume of water used for the infusion (coffee to water ratio) [6]. AA levels were highly dependent on the degree of coffee roasting as well; specifically, higher levels of AA were produced at medium and light degrees while increasing the degree of roasting rapidly reduced the acrylamide content [7]. Recently a study reported that Robusta coffee beans had a higher AA content than Arabica, which was due to the higher asparagine content (limiting factor) [8].

### 1.2. Antioxidant Compounds in Coffee

In addition, nowadays, coffee is known as a beverage rich in bioactive substances which potentially possess multiple effects on the body and human health. Polyphenols, such as chlorogenic acids (CGAs), are the best-known bioactive compounds in coffee beverages [9] and are associated with antioxidant properties and some health benefits have been attributed to them, e.g., prevention of chronic diseases such as cancer and cardiovascular disease [10]. Their composition in coffee beverages varies and depends on many factors (species of coffee, roasting of coffee beans, infusion conditions, etc.) [9]. Some studies [11] have examined the effects of various antioxidants, like CGAs, on AA formation, but with conflicting results. In some of them, a decrease or increase in AA content was observed, while in others no changes were reported. These contrasting results may be due to different interactions occurring between functional groups of antioxidants compounds and precursors or reaction intermediates of AA. The purpose of this study was to investigate, for the first time, which infusion/extraction parameters had the greatest influence on the final content of both unhealthy substances, such as AA, and healthy substances, such as CGAs and antioxidant power, in diverse coffee brews prepared with different methods and using three bean varieties. To date, numerous studies on coffee preparation procedures have been reported in the literature, but rarely do they consider so many brewing preparations. Among the different methods used for specialty and filter coffee applications in recent times Turkish ibrik (boiling method), French press (steeping or immersion method), V60, Chemex, Clever (filtration or drip methods), AeroPress and Moka (pressure methods) have been the most proposed. To give a comprehensive overview of the filter coffee world, in this study the new Pure Brew (Victoria Arduino) technique was compared with these seven extraction methods. Therefore, the present work aimed to also investigate the differences between the new filter coffee extraction method, Pure Brew, and traditional ones (Turkish Ibrik, French Press, V60, Chemex, Clever, AeroPress and Moka), in terms of coffee extraction yield, AA content and antioxidant activity, with the final aim of establishing which parameters can affect the extraction of healthy and unhealthy substances more. These data, together with those from the spectrophotometric tests (DPPH, TFC and TPC), were finally elaborated via a statistical tool.

## 2. Experimental Section

### 2.1. Coffee Samples

Three different coffees with varying degrees of roast were applied for each of the eight extraction methods: Gardelli Specialty’s natural, non-classic anaerobic Ethiopia Uraga for a *light roast*, Gardelli Specialty’s washed Kenya Thiriku for a *medium roast*, and roasted Starbucks Blond 100% Arabica for a *dark roast*. These coffees were selected to provide a realistic picture of the world of “filter” and “specialty” coffee. As reported by [12], consumers base their choice mainly on the degree of roast that most satisfies them and do not select the item based on the type of green coffee.

All of the coffee samples were prepared by professional baristas and the coffee beans were ground with a professional grinder (Atom Brew Pro—Eureka, Florence). In the preparation of coffee beverages, water is also an essential ingredient, and its ionic content is fundamental for the preparation of coffee [13]. The same commercial brand was used for the preparation of all coffee samples, namely Nerea water. The chemical composition of the water was indeed suitable to be an effective extractant: according to the SCA parameters, the water must have a dry residue in the range of 75–250 mg/L and an ideal level of 100–150 mg/L. This range ensures that the water contains enough minerals for proper extraction without overwhelming the coffee with excessive minerals. The presence of minerals in the water was assessed by the producer (Nerea, Castelsantangelo sul Nera, MC, Italy). The magnesium concentration was 0.9 mg/L, and the sodium concentration was 1.18 mg/L. The sodium content should ideally be below 10 mg/L. Excessive sodium can lead to a salty taste. Regarding water hardness, this refers to the concentration of calcium ions. In coffee preparation, extremely soft water can result in under extraction and flat flavors, while very hard water might lead to over extraction and bitterness, so moderately hard water is generally considered ideal for coffee brewing. The SCA suggests a calcium hardness level of 50–175 ppm. The water used in this study reported a calcium concentration of 58 mg/L.

### 2.2. Chemical and Reagents

AA (for molecular biology ≥ 99% (HPLC), C_3_H_5_NO, molecular weight 71.08 g/mol, CAS No 79–06-1) and2,3,3-d_3_-acrylamide (AA-d_3_) standard solution, 500 mg/mL in acetonitrile (analytical standard, CAS 122775-19-3) was purchased from Sigma Aldrich (St. Louis, MO, USA). The individual stock solution of AA was prepared by dissolving the pure standard compound in ultra-pure water at a concentration of 1000 mg/L and stored in glass-stoppered bottles at −18 °C. Afterwards, standard working solutions at various concentrations were prepared daily by appropriate dilution of the stock solution with water. A solution of AA-d_3_ was combined with standard working solutions of native AA prepared at various concentrations in order to obtain a concentration of 500 ng/mL [7]. HPLC-grade acetonitrile and methanol were supplied by Sigma-Aldrich (Milano, Italy). HPLC-grade formic acid (99%) was obtained from Merck (Darmstadt, Germany). Bond Elut-Accucat, 200 mg, 3 mL cartridges for solid-phase extraction (SPE) were bought from Agilent Technology (Santa Clara, CA, USA), while Oasis HLB 200 mg, 6 mL cartridges were purchased from Waters (Milford, MA, USA). Deionized water was further purified using a Milli-Q SP Reagent Water System (Millipore, Bedford, MA, USA). Before high-performance liquid chromatography–tandem mass (HPLC-MS/MS) analysis, all samples were filtered with a Phenex™ RC 4 mm 0.2 μm syringeless filter. The Folin–Ciocolteu reagent, sodium carbonate (Na_2_CO_3_), gallic acid (C_7_H_6_O_5_), TPTZ (2,4,6-tri(2-pyridyl)-S-triazine), ferric chloride hexahydrate (FeCl_3_⋅6H_2_O), sodium acetate (C_2_H_3_O_2_Na), acetic acid (C_2_H_4_O_2_), trolox (6-hydroxy-2,5,7,8-tetramethylchroman-2-carboxylic acid), ABTS (2,2′-azinobis(3-ethylbenzothiazoline-6-sulfonic acid) diammonium salt), potassium persulfate (K_2_S_2_O_8_), disodium phosphate (Na_2_HPO_4_), monopotassium phosphate (KH_2_PO_4_), sodium acetate (C_2_H_3_O_2_Na) and ethanol (C_2_H_5_OH) were purchased from Sigma-Aldrich (St. Louis, MO, USA). The hydrogen chloride (HCl), potassium chloride (KCl), acetic acid (CH_3_COOH), sodium hydroxide (NaOH) and glycerin (C_3_H_8_O_3_) were acquired from Carlo Erba reagents (Milan, Italy). The DPPH (2,2-diphenyl-1-picrylhydrazyl) was obtained from Glentham Life Sciences (Corsham, UK). All chemicals and reagents were analytical grade.

### 2.3. Coffee Brew Preparation Methods

A specific routine was used for each of the eight preparation methods, keeping some parameters as constant as possible, but without distorting the beverage recipes. Three replicates were made for each brewing method. The data are reported in Table 1.

#### 2.3.1. AeroPress

The AeroPress was invented by Aerobie (Alan Adler) in 2005. The device consists of two nested cylinders; one has a flexible, hermetic seal and “stays” inside the larger cylinder [14]. The paper filter (AeroPress^®^ Micro Filter, Palo Alto, USA) was first moistened with water and placed on the bottom of the AeroPress syringe, then the device was placed on a scale and a tare was made; the coffee was repoured, and the tare was made again. Then, the blooming phase (the phase in which all the ground coffee is covered with water) was performed three times; 60 mL of water was poured to wet all the coffee and it was mixed three times with a spatula; 30 s later, the remaining water was poured all at once; then the slurry was mixed three times with the spatula to achieve optimal extraction. Finally, the lid of the AeroPress was opened by applying pressure for about 40 s. In this case, the coffee: water infusion ratio used was 15:225.

#### 2.3.2. Chemex

Chemex is a variant of filter coffee that originated in the United States in the early 1940s (Chemex Coffee Maker 1941). The Chemex carafe is made of transparent glass and has a typical hourglass shape. The inner surface of the cone in which the filter (Chemex bonded Filter, Natural squares, Chicopee, MA) is inserted is completely smooth and without grooves. This allows the filter to retain more coffee particles. During preparation, the filter was inserted into the opening of the carafe. The filter was first moistened with warm, not boiling, water, which was then discarded. Hot water (93 °C) was poured over the ground coffee a total of two times, allowing about 30 s to pass, with circular motions made and mixed after each addition of water. Then, the coffee powder was added to the filter with a brewing ratio of 20:300 (coffee:water).

#### 2.3.3. Clever

The Clever Dripper is a 2008 invention by the Taiwanese company Abid Co [15]. The Clever consists of a plastic cone with a valve, when the valve is closed, the ground coffee is infused in water, while when the valve is open, percolation takes place. First, 300 g of water at a temperature of 93 °C was poured in, and then the ground coffee was mixed for 10 s with a spatula. Finally, about 3 min after the start of the infusion, the mixture was filtered. The filter used was the Filtertuten original, Filtertraditional (Melitta, Clearwater, USA). A brewing ratio of 20:300, was selected for the preparation of the Clever.

#### 2.3.4. French Press

The French Press (Lacor French Press wood) consists of a cylindrical pot with a plunger that forms a knob and ends with a metal mesh filter [16,17]. The lid is used to close the coffee maker. At the beginning, the coffee was ground and put into the device. Then, the device was calibrated, and water was added. During this last process, strong turbulence was generated from the top at 1 min, 2 min and 3 min, then the lid was removed, and the liquid phase was rotated four times with the spatula provided. After 4 min, the coffee was filtered, the cap was replaced, and the filter was slowly pressed into the coffee. The ratio of coffee to water was 20:300.

#### 2.3.5. Moka

Moka is one of the most used methods of coffee preparation in Italy and was invented by Alfonso Bialetti in 1933 [18]. The device consists of a container equipped with a valve that contains the water, above it there is a funnel-shaped filter into which the ground coffee is placed, and finally, in the upper part, the collecting container from which the finished coffee is expelled. The water contained in the water heater and exposed to heat reaches a pressure of about 1−2 atm, which allows it to climb back up the filter and finally enter the collection container. A Bialetti Moka Express for 6 cups was used for the preparation. For the brewing procedure, 250 g of water and 25 g of ground coffee were used.

#### 2.3.6. Pure Brew

The Pure Brew is obtained with the VA388 Black Eagle Maverick machine (Simonelli Group, Victoria Arduino). Pure Brew technology is an extraction method that uses pulsating frequencies using low-pressure water (less than 0.15 bar). The Pure Brew coffee filter consists of a micro-thin double-mesh conical basket that can contain up to 20 g of coffee [12,16]. Combining Pure Brew technology with the patented filter basket made it possible to obtain filtered coffee by pressing a button. The water temperature was 93 °C. The coffee-water ratio was 20:260.

#### 2.3.7. Ibrik Coffee (Turkish Coffee)

Turkish coffee is prepared with the ibrik, a small brass vessel very common in the East. Sometimes it is enriched with spices. For the preparation, 8 g of coffee was placed into the ibrik with 80 g of water at 70 °C (first the coffee, then the water), mixed and the infusion was brought to a boil twice. For the preparation, a brewing ratio of 1:10 coffee: water was chosen.

#### 2.3.8. V60

This coffee maker consists of three parts: a cone-shaped upper dripper with ribs along the inner edges and a single large hole in the lower part, a paper filter (CAFEC filter, white paper, Oita, Japan) and a glass container (Hario V60 Range Server 600 mL) [14]. First, a small amount of 93 °C hot water was poured to wet the filter, then the coffee was placed in until a flat surface was reached. Then, 60 mL of 93 °C hot water was poured on top of the coffee, which was pre-infused for 15 s; the water was always poured in concentric circles, starting in the center, and then moving outward, trying to maintain a constant flow; another 100 mL of water was poured after 30 s. The water was poured into the coffee filter. The brewing ratio was 20:300 coffee: water.

### 2.4. Brewing Characteristics (pH, TDS % and EY)

To determine the infusion properties, pH and total dissolved solids (TDS %) were measured and the extraction yield (EY) was calculated. The pH measurement was performed using a digital pH meter (Mettler Toledo, Columbus, UK). Extraction yield was calculated from TDS % values measured with a refractometer (VST LAB Coffee III Refractometer, Rancho CA, USA). The TDS % value is considered the strength of the brew, i.e., the mass fraction of soluble solids in the brew, while the extraction percentage (PE) is expressed by the “extraction yield”, i.e., the mass fraction of soluble solids removed from the coffee grounds [19]. These values were included by Lockhart in the classic “Coffee Brewing Control Chart”. This chart serves as the basis for professional training in the coffee industry and is the basis for the stringent requirements for certification of home brewers [16]. The chart is divided into nine regions, with vertical separation versus TDS % values labeled as “strong” or “weak” and horizontal separation versus PE labeled as “bitter” or “underdeveloped”. The data are shown in Table 1.

### 2.5. Acrylamide Extraction and Sample Clean-Up

For the extraction of AA and purification of samples, a previous procedure was used [20,21] with some adaptations. To start, 3 mL of the coffee sample was shaken with a vortex mixer for 30 s, and after centrifugation at 5000 rpm for 10 min, 1.5 mL of the supernatant was collected and 1 mL of the internal STD (containing 500 ng/mL) was added, then the sample was filtered using a 0.45 μm filter and purified using two different SPE cartridges. The first cartridge was the Oasis HLB. This was first conditioned with 3.5 mL of methanol and then with 3.5 mL of water. Then, 1.5 mL of the sample was loaded onto the cartridge followed by 0.5 mL of water. The sample was allowed to pass completely through the sorbent material. For the AA elution, water (1.5 mL) was added to the cartridge and the eluent was collected in a 3 mL glass vial. Prior to conditioning the second SPE column, a marker was placed on the outside of the cartridge at a height corresponding to 1 mL of liquid above the sorbent bed. The Bond Elut-Accucat column was conditioned with 2.5 mL of methanol followed by 2.5 mL of water. The solvents used for conditioning were then discarded. The eluent collected from the first cartridge was added to the Bond Elut-Accucat cartridge. The sample was eluted from the column to the mark previously made on the outside; the eluent was then collected in a 6 mL glass vial. Finally, samples were filtered with a 0.2 μm filter and injected into the HPLC-MS/MS.

### 2.6. Particle Size Analysis

For each extraction method, the baristas chose a specific grain size of coffee powder to be able to obtain the desired result. A portion of the coffee powder was analyzed with the Mastersizer 3000 Aero Series (Malvern PANalytical Ltd., Malvern, UK) equipped with a dry dispersion unit. This instrument provided a particle size measurement from 0.01 to 3500 μm. The Mastersizer 3000 Aero Series works by the diffraction of laser beams that interact with the ground coffee particles. The device worked with a non-stop air flow, generated by a mechanical compressor at 6.5 bar. It transferred the particles at 2–3 bar to the ray diffraction. In this way, the particles moved in laminar flow and the vacuum extraction unit (KARCHER Professional NT 45/1 Tact, Munich, Germany) removed the samples from the aero dry. Ground coffee powders with various particle sizes were collected: one-fifth for Mastersizer 3000 and the rest for the extraction of filter coffees. The particle size distribution for each sample was checked five times and the mean value was applied for comparison. The results are reported in Appendix A and are presented in the form of a histogram in Figure 1.

### 2.7. Acrylamide Analysis by HPLC-MS/MS System

The quantification of acrylamide in different samples of filter coffee was performed according to a previously developed and validated method [22]. The instruments used were an Agilent 1290 Infinity Series and a 6420 Triple Quadrupole from Agilent Technology (Santa Clara, CA, USA) equipped with an electrospray ionization (ESI) source set up in positive polarity. The column used for the present analytical method was a KinetexHilic (100 mm × 4.6 mm i.d., particle size 2.6 μm) from Phenomenex (Torrance, CA, USA), preceded by a KrudKatcher ULTRA HPLC in-line filter (depth filter 2.0 μm × 0.004 in i.d.). The mobile phase consisted of water (A) and acetonitrile (B), both containing 0.1% formic acid, and separation was performed at 0.8 mL min^−1^ in gradient elution mode. The mobile phase ranged as follows: 0–2.5 min, 85% B; 2.5–3.5 min, 85–70% B; 3.5–5.5 min, 70% B; 5.5–6.5 min, 70–60% B; 6.5–10 min, 60% B. The injection volume was 2 μL, the temperature of the drying gas in the source was maintained at 350 °C, the gas flow was 12 L min^−1^, the nebulizer pressure was 45 psi, and the capillary voltage was set at 4000 V. The acquisition was performed in “Multiple Reaction Monitoring” (MRM) mode.

### 2.8. Chlorogenic Acids and Caffeine Analysis

Analysis of chlorogenic acids and caffeine was performed according to a previously developed and validated method by Santanatoglia et al., 2023 [23]; using an Agilent 1100 (Agilent Technologies, Santa Clara, CA, USA) consisting of a diode array detector (DAD), a binary pump, and an autosampler. The injection volume was 3 μL, and the analytical column used was a Gemini C18 (250 mm × 3.0 mm, 5 μm) preceded by a Security Guard C18 column (4 cm × 3 mm, 5 μm) (Phenomenex, Torrance, CA, USA). The column temperature was set at 40 °C, and elution was performed in gradient mode using water (A) and methanol (B), both with 0.1% formic acid as the mobile phase. The wavelength used was always 325 nm for chlorogenic acids and 270 nm for caffeine.

### 2.9. Spectrophotometric Analysis

#### 2.9.1. DPPH

Antioxidant activity was determined using the DPPH method, spectrophotometrically evaluating the decay of the 2,2-DiPhenyl-1-Picril-Hidrazyl radical by antioxidant substances. These tests were carried out on the filter coffees after they were diluted 1:25 in water, following the procedure described by Abouelenein et al., 2023 [24], with some modifications. Briefly, 0.5 mL of the diluted coffee was mixed with 4.5 mL of an ethanolic DPPH solution (0.1 mM). After 30 min of incubation in the dark at room temperature, the decrease in the DPPH radical was measured spectrophotometrically at 517 nm using an Agilent Cary 8454 UV-Vis spectrophotometer. Trolox was used as the reference antioxidant, and the results were expressed as mg Trolox equivalent (TE)/L coffee beverage.

#### 2.9.2. Determination of Total Phenolic Content (TPC) and Total Flavonoid Content (TFC)

TPC was determined spectrophotometrically according to the method Abouelenein et al., 2023 [24], with some modifications. All coffee samples were diluted 1:25 in water. Briefly, 0.5 mL of the sample solution was added to the tubes, then 2.5 mL of the Folin–Denis reagent solution and 7 mL of Na_2_CO_3_ solution (7.5% *w*/*w* in water) were added. The reaction mixture was allowed to stand in the dark at room temperature for 2 h, and the absorbance was measured at 765 nm. Quantification of TPC in the extracts was performed using a gallic acid calibration curve and was expressed as mg gallic acid equivalents (GAE) per g of coffee beverage. The TFC of the different extracts was determined according to a method described by Abouelenein et al., 2023 [24], with slight variations. All coffee samples were diluted 1:25 in water. In a 15 mL tube, 0.5 mL of the sample solution, 0.15 mL NaNO_2_ (0.5 M), 3.2 mL methanol (30% *v*/*v*), and 0.15 mL AlCl_3_ 6H_2_O (0.3 M) were mixed. After 5 min, 1 mL of NaOH (1 M) was added. The solution was mixed well, and the absorbance was measured at 506 nm compared to the blank. The standard calibration curve for TFC was prepared using a rutin standard solution according to the same procedure as described above. TFC was expressed as mg rutin equivalents (RE) per g of coffee beverage.

### 2.10. Statistical Analysis

All analytical measurements on the coffee samples were performed in triplicate, and the results obtained were subjected to statistical analysis. Discriminant analysis based on the partial least squares method and the Pearson correlation were performed using MetaboAnalyst 5.0 tool (https://www.metaboanalyst.ca/ accessed on 25 March 2023). Therefore, partial least squares-discriminant analysis (PLS-DA) was performed to evaluate the discrimination between the groups of different filter coffee samples [12]. The Pearson correlation measures the linear relationship between two continuous variables and is commonly used to assess the strength and direction of the association between them; in this study it was applied to find positive or negative correlation among different analytes and extraction parameters.

## 3. Results and Discussion

### 3.1. Acrylamide Results among the Different Coffee Extraction Methods

The results of chromatographic analysis (HPLC-MS/MS) on AA content in the coffee samples are listed in Table 2 and presented in the form of histograms in Appendix A. The results in Table 2 are expressed as mean ± standard deviation of acrylamide (ng/mL). These results showed that the Moka coffee samples had the highest acrylamide levels compared to the other extraction methods for two roasts: 174.683 ± 32.33 ng/mL for the light roast and 170.060 ± 42.05 ng/mL for the dark roast. Only the beverage extracted by the French press method for the medium roast had higher acrylamide levels (172.256 ± 6.51 ng/mL), while the acrylamide levels of the AeroPress (107.854 ± 5.57 ng/mL for light, 56.724 ± 2.43 ng/mL for medium, and 37.793 ± 2.72 ng/mL for dark roast), Clever (45.812 ± 2.31 ng/mL for light, 47.681 ± 1.23 ng/mL for medium and 24.730 ± 2.97 ng/mL for dark roast) and Pure Brew (62.344 ± 5.82 ng/mL for light, 67.657 ± 2.57 ng/mL for medium and 29.723 ± 1.39 ng/mL for dark roast) appeared lower compared to the average of the three different coffees at different roast levels. Comparing the AA content reported in Table 2 with the brewing parameters reported in Table 1, it can be observed that the AA levels were the highest in the drinks that had a higher TDS % and a higher brew ratio (this refers to the relationship between the amount of coffee grounds and the amount of water used during the brewing process). Conversely, they were lower where the TDS % and the brew ratio were lower. This was observed for all three coffees with different roasting degrees, as reported by [4,5], who found that acrylamide concentrations decreased simultaneously with a decrease in coffee to water ratio: from 108.2 to 50.1 ng/mL and from 56.3 to 24.2 ng/mL, respectively. Moreover, it is possible to observe a progressive decrease in AA levels in coffee samples from light to dark roast coffee. The average levels of acrylamide in coffee beverages prepared with light roasted coffee were 93.273 ng/mL, those of beverages obtained with medium roast were slightly lower, more precisely 90.202 ng/mL, and finally the lowest levels of AA were obtained in coffee beverages prepared with dark roasted coffee with a content of 53.105 ng/mL. This trend with a decrease in AA levels as the roasting of the coffee beans increases is an effect already known in the literature and observed by [22,25].

### 3.2. Chlorogenic Acids and Caffeine Analysis Results on Filter Coffee Extraction Methods

Table 3 and Table 4 show the results of HPLC-DAD analysis of chlorogenic acids and caffeine, and Appendix A shows the corresponding histograms. From these results, Moka and Turkish coffee contained higher levels of total chlorogenic acids than other extraction methods in each different roasted coffee. In terms of total of CGAs detected: the Moka and Turkish had 5554.74 and 5598.54 mg/L, for light, 4207.48 and 4217.80 mg/L for medium, 2568.76 and 2322.89 mg/L for dark, respectively. The total chlorogenic acid content was in the medium range for AeroPress (2549.02 mg/L for light, 1648.94 mg/L for medium and 1411.06 mg/L for dark roast), while lower contents were found in Pure Brew (2714.14 for light, 1530.50 for medium and 1353.95 mg/L for dark roast), Chemex (2438.88 mg/L for light, 1408.14 mg/L for medium and 1179.23 mg/L for dark roast) and Clever (2223.48 for light, 1533.93 for medium and 1294.80 mg/L for dark roast).

Caffeine levels showed a similar trend in Table 4, with the Moka and Turkish coffee samples having the highest levels of this analyte (2903.14 ± 26.69 mg/L and 2055.95 ± 128.31 mg/L for light, 2661.53 ± 66.72 mg/L and 1955.33 ± 13.98 mg/L for medium, 2342.32 ± 86.68 mg/L and 2551.03 ± 17.61 mg/L for dark roast, respectively), while Pure Brew (1188.00 ± 20.82 mg/L for light, 1173.88 ± 10.25 mg/L for medium, and 1391.57 ± 2.35 mg/L for dark roast), Chemex (1056.74 ± 1.07 mg/L for light, 727.26 ± 3.52 mg/L for medium, and 1251.10 ± 8.01 mg/L for dark roast), and others, displayed lower contents of caffeine. In addition, the Moka and Turkish coffees had higher brew ratios (12.35 and 10.00 for the light, 12.39 and 10.00 for the medium, and 12.36 and 10.00 for the dark) and EY (21.13 and 26.1 for the light, 18.88 and 26.8 for the medium, and 22.49 and 28 for the dark) than all the other studied preparation methods. These observed caffeine analysis results were comparable to the values reported by Angeloni et al., 2019 [14] (740 mg/L for V60 and 1280 mg/L for Moka). These results confirmed the observation of Andueza, Vila, Paz de Peña and Cid, 2007 [26] that the extraction of caffeine and chlorogenic acid, compounds associated with bitterness and astringency, increases when the coffee-water ratio is higher.

### 3.3. Spectrophotometric Analysis Results on Filter Coffee Extraction Methods

Table 5 shows the results of the spectrophotometric analyses, reporting data on DPPH, TFC, and TPC.

#### 3.3.1. Antioxidant Activity, DPPH Assay

From the results of the DPPH tests (Table 5), for light and medium roasted coffee, the V60 coffee and Pure Brew had significantly lower values than the others (4216.42 ± 407.83 and 4515.62 ± 64.02 mg Trolox equivalents/L of coffee beverage for light, 3819.71 ± 91.47 and 4414.46 ± 84.28 mg Trolox equivalents/L of coffee beverage for medium roast, respectively). For dark roasted coffee samples, DPPH test results were lower in the V60 coffee samples (4920.68 ± 194.84 mg Trolox equivalents/L of coffee beverage) and Clever coffee (4997.87 ± 373.69 mg Trolox equivalents/L of coffee beverage). The highest results of the DPPH tests were obtained for the AeroPress (5569.80 ± 15.35 mg Trolox equivalents/L of coffee beverage) and Clever (5584.53 ± 4.38 mg Trolox equivalents/L of coffee beverage) for light roasts, for the Moka (5438.54 ± 15.14 mg Trolox equivalents/L of coffee beverage) and Turkish coffee (5345.52 ± 3.98 mg Trolox equivalents/L of coffee beverage) for medium roasts, and for the AeroPress (5330.78 ± 9.97 mg Trolox equivalents/L of coffee beverage) and Moka (5320.87 ± 14.97 mg Trolox equivalents/L of coffee beverage) for dark roasts.

#### 3.3.2. Total Flavonoid Content (TFC)

From the results of the TFC spectrophotometer test (Table 5), higher flavonoid contents were found in the Moka and Turkish coffee samples for each of the three different roasts (respectively, 10.08 ± 2.02 and 12.01 ± 2.06 mg rutin equivalents/g of coffee beverage for light roasts, 8.72 ± 0.02 and 11.06 ± 0.61 mg rutin equivalents/g of coffee beverage for medium roasts, 8.58 ± 1.26 and 8.23 ± 1.59 mg rutin equivalents/g of coffee beverage for dark roasts). The AeroPress (3.80 ± 0.56 mg rutin equivalents/g of coffee beverage) for light, Clever (2.01 ± 0.74 mg rutin equivalents/g of coffee beverage) for medium, and V60 (2.96 ± 0.50 mg rutin equivalents/g of coffee beverage) for dark roasts displayed the lowest flavonoid contents. The other extraction methods showed medium flavonoid contents. These results of the TFC analysis were comparable to the values reported by Gobbi, Maddaloni, Prencipe and Vinci (2023) [26] in the literature (8.50–8.60 mg RUT/g of coffee).

#### 3.3.3. Total Phenolic Content (TPC)

From the results of the TPC spectrophotometric tests (Table 5), higher levels of phenolic compounds were observed in the Moka and Turkish coffee samples for each of the three different roasts (respectively 4.68 ± 0.14 and 4.45 ± 0.72 mg gallic acid equivalents/g of coffee beverage for light roasts, 3.30 ± 0.21 and 3.51 ± 0.04 mg gallic acid equivalents/g of coffee beverage for medium roasts, 4.04 ± 0.11 and 5.04 ± 0.19 mg gallic acid equivalents/g of coffee beverage for dark roasts). The lowest levels of phenolic compounds were found for the French Press (2.08 ± 0.12 mg gallic acid equivalents/g of coffee beverage) for light roasts, the AeroPress (1.73 ± 0.16 mg gallic acid equivalents/g of coffee beverage) for medium roasts, and the V60 (2.15 ± 0.24 mg gallic acid equivalents/g of coffee beverage) for dark roasts. The other extraction methods had intermediate levels of phenolic compounds. These results of the TPC analysis were comparable to the values reported by Gobbi, Maddaloni, Prencipe and Vinci (2023) [26] (2.71–3.52 mg GAE/g of coffee powder). These results, compared to the data listed in Table 1, show a potential association: the levels obtained from the analysis of the antioxidant power seem to be associated with some infusion parameters such as the coffee-water ratio, EY and the TDS % values. More precisely, an increase in antioxidant power was observed with the increase in these parameters, particularly with the brew ratio and with the EY. On the other hand, lower values of TDS %, brew ratio and EY were observed as the antioxidant power of the analyzed coffee samples decreased [27]. Hence, the Pearson correlation was applied to study correlations among each of the analyzed compounds and the investigated extraction parameters.

### 3.4. Statistical Analysis

#### 3.4.1. Multivariate Statistical Discrimination of the Different Extraction Methods

The analysis of the filter beverages prepared with three different coffees allowed the quantification of some well-known coffee compounds important for flavor, and they also possess some health and unhealth activities. A chemometric tool has been applied to potential discriminate the coffee extraction methods considering the presence of healthy and unhealthy molecules as well as the investigated physiochemical parameters. The partial least squared-discriminant analysis (PLS-DA) score plot revealed a quite good discrimination in this sense (Figure 2A). Among the three types of coffee (light, medium and dark roast), the Turkish and Moka systems stand out from the others. This could be because these two extraction types had higher concentrations of analytes than other extraction methods for most of the chemical analyses performed. In addition, both Moka and Turkish systems differed by having higher TDS % and brew ratio values, as shown in Table 1. The Pure Brew method was close to the classic extraction methods, except for light roast coffee, where it was found to be slightly different. For light roast coffee, the score plot in Figure 2A shows that the Turkish and Moka were close, as were Chemex and Clever, probably this could be due to the filter and its similar shape, while similarity between the French Press and AeroPress system, could probably be because they are systems that use pressure. On the other hand, the V60 and Pure Brew were further apart than the traditional extraction methods; this could be attributed to the fact that they were pour-over systems. The score plot shown in Figure 2A, reporting data on medium roast coffee, shows that the Turkish and Moka systems were always separated from each other and from the other extraction methods while the Pure Brew, French Press, AeroPress and Clever extraction systems were closely grouped, and the Chemex and V60 were slightly separated. Also, in the case of dark roasted coffee (the mainstream of Starbucks), the Turkish and Moka were very close, almost overlapping, and clearly separated from the other extraction methods. Notably, the Chemex was slightly separated from the traditional extraction methods (Pure Brew, Clever, AeroPress, French Press and V60) which were instead clearly clustered. In conclusion, it was possible to find a recurring separation of the Moka and Turkish systems, which was probably attributable to the variables with the highest VIP scores. In fact, in the PLS-DA model, scores of VIP estimated the importance of each variable in PLS projection [28]. Figure 2B represents the top 10 variables with higher VIP scores, which fluctuated wildly among the samples. In particular, two of them reported a VIP score higher than 1.0, meaning these are important factors for the discrimination of filter coffee extraction methods based on the considered parameters. These parameters belong to chemical classes of antioxidant molecules, being the CGAs compounds and TFC assays results. The main influencing VIP was the total value of CGAs.

#### 3.4.2. Pearson Correlation

Figure 3 shows the Pearson correlations calculated between different coffee parameters and analyte content in the three types of coffee, i.e., light, medium and dark. The data shown from the three plots were in accordance with those reported in the literature by Von Blittersdorff and Klatt, 2017 [29], and Cordoba, Fernandez-Alduenda, Moreno, and Ruiz, 2020 [30]. Specifically, it has been demonstrated that small and irregular particles released their soluble substances faster and led to a more intense, concentrated coffee in the cup. In fact, especially in the cases of washed coffee for the medium and dark roasted coffee types, strong negative correlations (in blue in the plots) between the particle size and the levels of AA, total CQAs, total phenolic compounds content, caffeine, and DPPH, were found. Therefore, the significant influence of the size of coffee particles in the extraction process was shown, as has already been reported. In addition, for each of the coffee types (light, medium and dark roast), another important positive correlation (darker red color plots) between the coffee to water ratio and the contents of AA, total CQA, total phenolic compounds, total flavonoids, caffeine, and DPPH were obtained. This was already reported by Andueza et al., 2007 [31] and Cordoba, Fernandez-Alduenda, Moreno, and Ruiz, 2020 [12], who found the direct effect of coffee to water ratio on the chemical properties of the coffee infusion, and they demonstrated that the use of a higher coffee to water ratio improved the extraction of caffeine and chlorogenic acids. Another similar trend, for each of the three coffee types (light, medium and dark roasted) was observed between the extraction time and the content of AA, total CQAs, total phenolic compounds, total flavonoids, caffeine and DPPH. This result was also confirmed in the literature where it was reported that a longer brewing time could lead to a higher extraction of certain compounds, e.g., some antioxidants [13,30]. 

## 4. Conclusions

The process of coffee extraction is very complex, with many variables directly affecting the chemical composition and the flavor of the coffee cup. For this reason, new processes were constantly studied and developed by the coffee industry. The results of this study confirmed that infusion parameters, such as particle size, brewing ratio, and extraction time, have a significant influence on the final content of healthy and unhealthy investigated substances, i.e., acrylamide, chlorogenic acids and caffeine. It was observed that the AA levels were the highest in the drinks that had a higher TDS % and a higher brew ratio, instead they were lower where the TDS % and the brew ratio were lower; this was observed for all three coffees with different degrees of roasting. Moreover, it was possible to observe a progressive decrease in AA levels in coffee samples from light to dark roast coffee. The average levels of acrylamide in coffee beverages prepared with light roasted coffee were 93.273 ng/mL, those of beverages obtained with a medium roast were slightly lower and more precisely 90.202 ng/mL, and finally the lowest levels of AA were obtained in coffee beverages prepared with dark roasted coffee with a content of 53.105 ng/mL. Further research into the impact of AA will be needed to determine safe levels in the coffee beverage. However, it is clear from these results that, in the final beverage, the level of this analyte was generally quite low. In addition, the variables with the most impact on the discrimination of the studied preparation methods were the total content of chlorogenic acids, TFC, TPC, caffeine and DPPH assay. For this reason, the recipe and infusion parameters used for each of the extraction systems are the key factors that regulate the extraction of coffee constituents and, therefore, the quality of the cup.

## Figures and Tables

**Figure 1 antioxidants-12-01888-f001:**
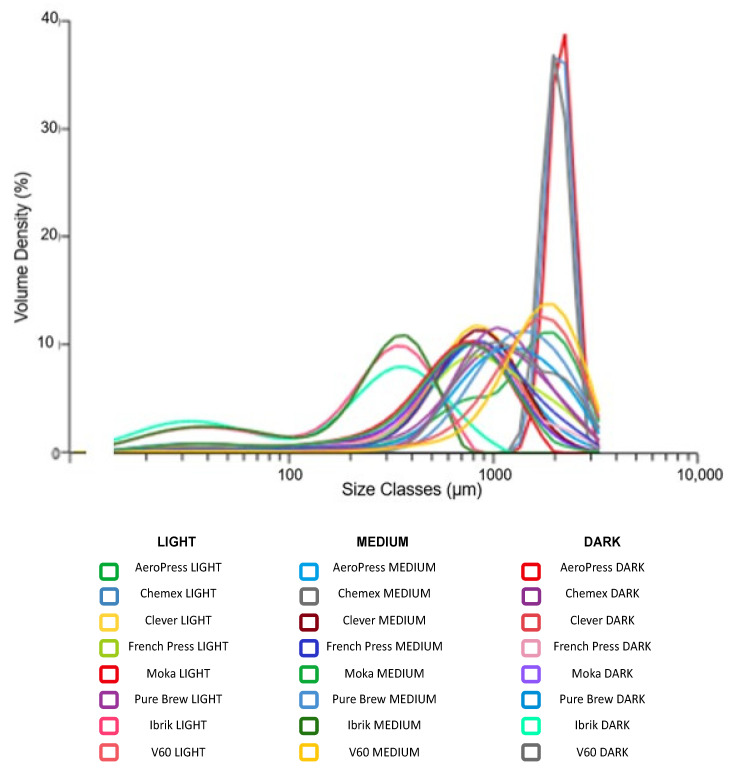
Granulometry, or the size of the coffee grounds, plays a crucial role in the extraction process and can affect the flavor and quality of the final cup of coffee. Different filter coffee extraction methods require specific grind sizes to achieve optimal results. Indeed, for each extraction method, a specific particle size of the coffee powder was chosen to obtain the desired result. The coffee powders were analyzed with the Mastersizer 3000 Aero Series. For the AeroPress, Chemex, Clever and V60 a medium-fine grind was maintained; for the French Press and Pure Brew, a coarse grind; while for the Moka and Ibrik a fine grind was chosen. The specific values are reported in Appendix A.

**Figure 2 antioxidants-12-01888-f002:**
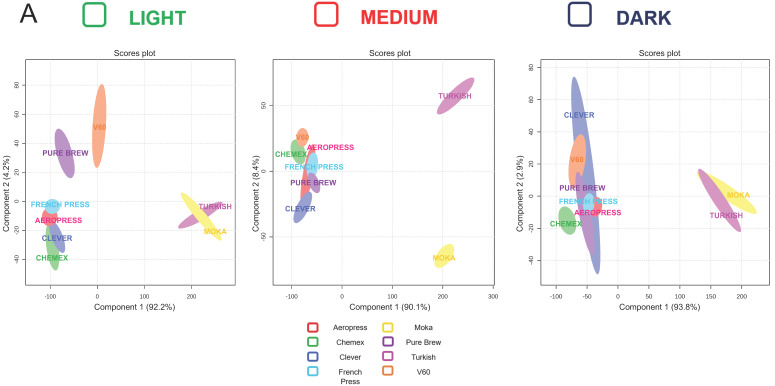
Discrimination of filter coffee samples according to extraction method. (**A**) Partial least squares-discriminant analysis (PLS-DA) scores plotted according to extraction method based on all parameters taken in consideration in the study. (**B**) Variable importance in projection (VIP) plotted values of the different analytes. The red line indicates the relevant variable cut-off (VIP score > 1).

**Figure 3 antioxidants-12-01888-f003:**
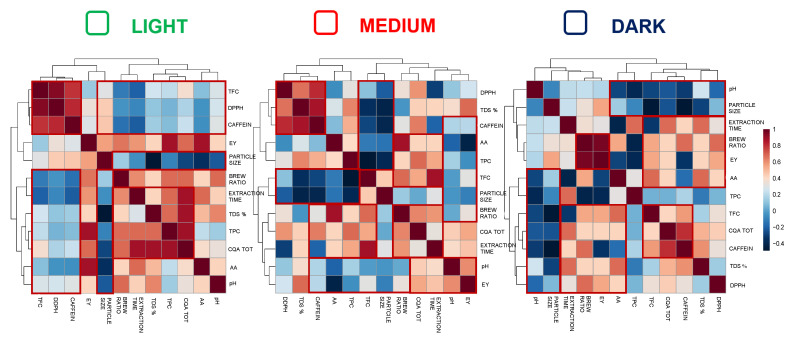
Pearson correlation coefficient of all the parameters considered in the study among the eight filter coffee samples, with the three differently roasted coffees. The Pearson correlation graphs can be used to identify patterns of positive or negative correlations between different extraction parameters, according to the numbers on the left.

**Table 1 antioxidants-12-01888-t001:** Extraction and resulting beverage parameters for each filter coffee: extraction method, amount of ground coffee, water temperature, extraction time, amount of coffee in the final cup, percentage of total dissolved solids (TDS %), beverage pH, extraction yield (EY) and brew ratio.

Extraction Methods	Coffee Powder (g)	Water Temperature (°C)	Time (s)	g Out	TDS %	pH	EY	Brew Ratio
AEROPRESS LIGHT	15	93	180 + 45	197.4 ± 0.8	1.37 ± 0.01	5.00 ± 0.02	18.02 ± 0.07	7.60
CLEVER LIGHT	20	93	265	254.6 ± 2.5	1.37 ± 0.02	4.99 ± 0.02	16.93 ± 0.18	7.76
CHEMEX LIGHT	20	93	155	257.7 ± 3.6	1.3 ± 0.01	5.00 ± 0.02	16.75 ± 0.21	7.86
FRENCHPRESS LIGHT	20	93	240	257 ± 0.3	1.36 ± 0.01	4.99 ± 0.02	17.47 ± 0.20	7.78
MOKA LIGHT	25	boiling	230	202.4 ± 0.6	2.61 ± 0.01	4.87 ± 0.04	21.13 ± 0.90	12.35
PURE BREW LIGHT	20	93	184	256.7 ± 0.2	1.29 ± 0.01	4.94 ± 0.03	16.55 ± 0.19	7.79
TURKISH LIGHT	8	boiling	150	80 ± 0.1	2.61 ± 0.01	4.84 ± 0.01	26.1 ± 0.14	10.00
V60 LIGHT	20	93	135	259.6 ± 2.4	1.35 ± 0.01	4.98 ± 0.01	17.52 ± 0.01	7.70

AEROPRESS MEDIUM	15	93	180 + 45	200.8 ± 0.4	1.38 ± 0.02	4.87 ± 0.01	18.47 ± 0.12	7.47
CLEVER MEDIUM	20	93	180 + 140	251.1 ± 2.3	1.35 ± 0.01	4.87 ± 0.02	16.94 ± 0.19	7.84
CHEMEX MEDIUM	20	93	260	255 ± 2.1	1.42 ± 0.01	4.88 ± 0.02	18.1 ± 0.14	7.97
FRENCHPRESS MEDIUM	20	93	240	259.2 ± 0.4	1.37 ± 0.01	4.85 ± 0.01	17.75 ± 0.20	7.71
MOKA MEDIUM	25	boiling	220	201.8 ± 0.2	2.34 ± 0.01	4.77 ± 0.03	18.88 ± 0.81	12.39
PURE BREW MEDIUM	20	93	187	262.3 ± 0.3	1.22 ± 0.01	4.80 ± 0.01	16 ± 0.14	7.62
TURKISH MEDIUM	8	boiling	145	80 ± 0.7	2.68 ± 0.01	4.79 ± 0.02	26.8 ± 0.04	10.00
V60 MEDIUM	20	93	200	259.8 ± 3.1	1.39 ± 0.01	4.87 ± 0.01	18.05 ± 0.02	7.70

AEROPRESS DARK	15	93	180 + 40	204.5 ± 3.1	1.45 ± 0.01	5.31 ± 0.01	19.76 ± 0.46	7.41
CLEVER DARK	20	93	270	253.3 ± 0.7	1.33 ± 0.02	5.32 ± 0.01	16.84 ± 0.21	7.65
CHEMEX DARK	20	93	193	261.3 ± 0.3	1.36 ± 0.01	5.27 ± 0.02	17.76 ± 0.16	7.91
FRENCHPRESS DARK	20	93	240	256.3 ± 0.2	1.44 ± 0.02	5.28 ± 0.02	17.44 ± 0.25	7.80
MOKA DARK	25	boiling	241	206.8 ± 3.1	2.72 ± 0.02	5.15 ± 0.01	22.49 ± 0.51	12.36
PURE BREW DARK	20	93	186	259.2 ± 0.4	1.46 ± 0.02	5.22 ± 0.01	18.92 ± 0.33	7.72
TURKISH DARK	8	boiling	130	80 ± 0.2	2.8 ± 0.01	5.15 ± 0.01	28 ± 0.07	10.00
V60 DARK	20	93	150	260.4 ± 1.2	1.35 ± 0.01	5.27 ± 0.01	17.57 ± 0.08	7.65

**Table 2 antioxidants-12-01888-t002:** Levels (mean ± standard deviation) of acrylamide (AA) in the analyzed filter coffee samples, obtained by HPLC-MS/MS analysis, expressed in ng/mL.

Sample	AA	Sample	AA	Sample	AA
AEROPRESS LIGHT	107.854 ± 5.57	AEROPRESS MEDIUM	56.724 ± 2.43	AEROPRESS DARK	37.793 ± 2.72
CHEMEX LIGHT	102.549 ± 1.13	CHEMEX MEDIUM	61.626 ± 2.97	CHEMEX DARK	44.652 ± 1.95
CLEVER LIGHT	45.812 ± 2.31	CLEVER MEDIUM	47.681 ± 1.23	CLEVER DARK	24.730 ± 2.98
FRENCH PRESS LIGHT	46.089 ± 1.52	FRENCH PRESS MEDIUM	172.256 ± 6.52	FRENCH PRESS DARK	28.449 ± 1.41
MOKA LIGHT	174.683 ± 32.33	MOKA MEDIUM	122.030 ± 2.43	MOKA DARK	170.060 ± 42.05
PURE BREW LIGHT	62.344 ± 5.82	PURE BREW MEDIUM	67.657 ± 2.58	PURE BREW DARK	29.723 ± 1.39
TURKISH LIGHT	79.375 ± 1.86	TURKISH MEDIUM	81.903 ± 0.005	TURKISH DARK	49.015 ± 3.71
V60 LIGHT	126.677 ± 25.70	V60 MEDIUM	111.741 ± 1.91	V60 DARK	40.417 ± 1.78
AVERAGE	93.123 ± 9.41		90.202 ± 2.51		53.105 ± 7.25

**Table 3 antioxidants-12-01888-t003:** Levels (mean ± standard deviation) of CGAs in the analyzed filter coffee samples, obtained by HPLC-DAD analysis, expressed in mg/L.

Sample	3-CQA	5-CQA	4-CQA	3,5-CQA
AEROPRESS LIGHT	482.6 ± 13.87	1389.24 ± 33.07	588.47 ± 13.88	88.71 ± 3.89
CHEMEX LIGHT	465.1 ± 15.55	1324.68 ± 50.14	565.68 ± 17.09	83.41 ± 11.91
CLEVER LIGHT	539.02 ± 53.00	1481.01 ± 11.33	124.32 ± 1.18	79.12 ± 0.59
FRENCH PRESS LIGHT	474.06 ± 107.86	1318.22 ± 43.73	568.46 ± 45.77	80.17 ± 32.08
MOKA LIGHT	1063.9 ± 78.18	2959.02 ± 185.83	1286.62 ± 141.21	245.17 ± 302.67
PURE BREW LIGHT	526.24 ± 80.91	1449.78 ± 231.89	623.49 ± 229.03	114.63 ± 170.21
TURKISH LIGHT	1080.21 ± 11.93	3011.89 ± 68.3	1292.25 ± 5.16	214.19 ± 37.39
V60 LIGHT	671.31 ± 3.45	1882.04 ± 26.07	807.88 ± 41.71	139.18 ± 78.32
AVERAGE	662.81 ± 45.59	1851.96 ± 81.29	732.15 ± 61.88	130.57 ± 79.63

AEROPRESS MEDIUM	331.01 ± 20.41	857.23 ± 38.65	392.38 ± 20.14	68.31 ± 19.46
CHEMEX MEDIUM	286.62 ± 58.04	720.07 ± 87.54	335.14 ± 97.64	66.40 ± 16.75
CLEVER MEDIUM	391.58 ± 3.18	984.37 ± 29.82	89.33 ± 3.13	68.65 ± 3.42
FRENCH PRESS MEDIUM	381.31 ± 18.90	965.57 ± 42.15	92.14 ± 0.85	58.08 ± 0.59
MOKA MEDIUM	850.15 ± 45.43	2161.08 ± 130.11	1005.68 ± 114.64	190.57 ± 52.72
PURE BREW MEDIUM	384.85 ± 35.07	987.52 ± 94.29	92.53 ± 27.67	65.61 ± 3.42
TURKISH MEDIUM	859.80 ± 203.11	2149.71 ± 577.23	1007.67 ± 425.07	200.61 ± 366.72
V60 MEDIUM	318.68 ± 5.92	810.59 ± 58.14	69.76 ± 4.06	53.81 ± 1.32
AVERAGE	475.5 ± 48.75	1204.52 ± 132.24	385.58 ± 86.65	96.505 ± 58.05

AEROPRESS DARK	314.14 ± 9.28	667.89 ± 22.49	371.76 ± 15.65	57.27 ± 3.49
CHEMEX DARK	278.02 ± 68.99	553.69 ± 120.94	316.74 ± 108.04	30.77 ± 36.57
CLEVER DARK	2863.56 ± 39.57	617.58 ± 66.22	347.27 ± 130.30	43.59 ± 91.89
FRENCH PRESS DARK	298.14 ± 80.03	611.98 ± 97.45	340.15 ± 174.54	49.41 ± 1.18
MOKA DARK	556.19 ± 30.19	1178.16 ± 90.54	676.02 ± 53.81	158.38 ± 11.09
PURE BREW DARK	305.14 ± 37.26	634.21 ± 83.13	355.43 ± 95.69	59.18 ± 81.39
TURKISH DARK	527.56 ± 8.13	1127.98 ± 43.31	609.39 ± 3.30	57.95 ± 79.15
V60 DARK	267.75 ± 58.65	591.21 ± 123.03	311.57 ± 102.46	36.68 ± 5.19
AVERAGE	676.31 ± 41.51	747.84 ± 80.89	416.04 ± 85.47	61.65 ± 38.75

**Table 4 antioxidants-12-01888-t004:** Levels (mean ± standard deviation) of caffeine in analyzed filter coffee samples, obtained by HPLC-DAD analysis, expressed in mg/L.

Sample	Caffein	Sample	Caffein	Sample	Caffein
AEROPRESS LIGHT	1065.49 ± 3.42	AEROPRESS MEDIUM	831.20 ± 0.64	AEROPRESS DARK	1587.29 ± 4.8
CHEMEX LIGHT	1056.74 ± 1.07	CHEMEX MEDIUM	727.26 ± 3.52	CHEMEX DARK	1251.10 ± 8.01
CLEVER LIGHT	1328.24 ± 1.39	CLEVER MEDIUM	1077.72 ± 12.38	CLEVER DARK	1430.82 ± 1.49
FRENCH PRESS LIGHT	992.88 ± 2.99	FRENCH PRESS MEDIUM	1086.70 ± 3.74	FRENCH PRESS DARK	1485.16 ± 1.92
MOKA LIGHT	2903.14 ± 26.69	MOKA MEDIUM	2661.53 ± 66.72	MOKA DARK	2342.32 ± 86.68
PURE BREW LIGHT	1188.00 ± 20.82	PURE BREW MEDIUM	1173.88 ± 10.25	PURE BREW DARK	1391.57 ± 2.35
TURKISH LIGHT	2055.95 ± 128.31	TURKISH MEDIUM	1955.33 ± 13.98	TURKISH DARK	2551.03 ± 17.61
V60 LIGHT	1532.94 ± 22.95	V60 MEDIUM	943.97 ± 6.19	V60 DARK	1306.73 ± 12.38
AVERAGE	1515.42 ± 25.96		1307.20 ± 14.64		1668.25 ± 16.91

**Table 5 antioxidants-12-01888-t005:** DPPH test, antioxidant activity (mg Trolox equivalents/L coffee beverage) in the analyzed filter coffee samples; total flavonoid content (TFC) (mg rutin equivalents/g coffee beverage) in the analyzed filter coffee samples; total phenolic content (TPC) (mg gallic acid equivalents/g coffee beverage) in the analyzed filter coffee samples.

Extraction Methods	DPPH, Antioxidant Activity mg Trolox Equivalents/L Coffee Beverage	TFC, Total Flavonoid Content mg Rutin Equivalents/g Coffee Beverage	TPC, Total Phenolic Content mg Gallic Acid Equivalents/g Coffee Beverage
AEROPRESS LIGHT	5569.80 ± 15.35	3.80 ± 0.56	2.36 ± 0.07
CHEMEX LIGHT	5532.46 ± 14.61	5.47 ± 0.82	3.43 ± 0.41
CLEVER LIGHT	5584.53 ± 4.38	6.12 ± 0.64	2.56 ± 0.05
FRENCH PRESS LIGHT	5265.57 ± 42.35	5.96 ± 0.03	2.08 ± 0.12
MOKA LIGHT	5403.56 ± 22.95	10.08 ± 2.02	4.68 ± 0.14
PURE BREW LIGHT	4515.62 ± 64.02	5.15 ± 1.56	2.12 ± 0.12
TURKISH LIGHT	5346.40 ± 0.54	12.01 ± 2.06	4.45 ± 0.72
V60 LIGHT	4216.42 ± 407.83	4.59 ± 0.64	2.68 ± 0.25
AVERAGE	5179.30 ± 71.50	6.65 ± 1.04	3.05 ± 0.20

AEROPRESS MEDIUM	4398.34 ± 73.68	4.53 ± 0.09	1.73 ± 0.16
CHEMEX MEDIUM	4006.38 ± 124.79	4.36 ± 0.04	2.11 ± 0.07
CLEVER MEDIUM	4226.88 ± 153.62	2.01 ± 0.74	1.97 ± 0.19
FRENCH PRESS MEDIUM	4369.99 ± 182.52	3.62 ± 0.01	2.28 ± 0.17
MOKA MEDIUM	5438.54 ± 15.14	8.72 ± 0.02	3.30 ± 0.21
PURE BREW MEDIUM	4414.46 ± 84.28	3.76 ± 0.29	1.90 ± 0.22
TURKISH MEDIUM	5345.52 ± 3.98	11.06 ± 0.61	3.51 ± 0.04
V60 MEDIUM	3819.71 ± 91.47	4.53 ± 0.28	2.34 ± 0.15
AVERAGE	4502.48 ± 91.19	5.32 ± 0.26	2.39 ± 0.15

AEROPRESS DARK	5330.78 ± 9.97	3.615 ± 0.07	2.80 ± 0.01
CHEMEX DARK	5273.92 ± 13.57	4.35 ± 0.80	2.46 ± 0.09
CLEVER DARK	4997.87 ± 373.69	3.98 ± 1.45	2.32 ± 0.07
FRENCH PRESS DARK	5232.16 ± 3.16	4.35 ± 0.17	2.85 ± 0.18
MOKA DARK	5320.87 ± 14.97	8.58 ± 1.26	4.04 ± 0.11
PURE BREW DARK	5281.69 ± 22.37	4.87 ± 1.34	2.45 ± 0.27
TURKISH DARK	5282.24 ± 17.52	8.23 ± 1.59	5.04 ± 0.19
V60 DARK	4920.68 ± 194.84	2.96 ± 0.50	2.15 ± 0.24
AVERAGE	5205.03 ± 81.26	5.12 ± 0.90	3.01 ± 0.15

## Data Availability

The data presented in this study are available in the article.

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
