# Peer review of "Effect of Brewing Methods on Acrylamide Content and Antioxidant Activity: Studying Eight Different Filter Coffee Preparations"

_antioxidants, 2023, doi:10.3390/antiox12101888_

Round 1

Reviewer 1 Report

1)      English should be improved e.g.: “contained higher contents”.

2)      Abstract: too general, no specific results are given.

3)      All Latin names of plant species and genera should be written Italic font.

4)      The references should be formatted according to the requirements of the Journal.

5)      Figure 1: the legend should be below the chart, and the text should be improved to be as informative as possible, it is not a scientific information that: “the baristas have chosen a specific particle size of the coffee 268 powder to obtain the desired result for each specific type of method …”.

6)      2.9.1. – why “mg Trolox equivalent (TE)/kg coffee beverage” –it should be expressed as mg TE/g dry weight of coffee beverage;

7)      “The results of chromatographic analysis (HPLC-MS/MS) on AA contents in the coffee samples are listed in Table 2A and presented in the form of histograms in Figure 1SA. The results in Table 2A are expressed as mean ± standard deviation of acrylamide  (ng/mL).” – this is unnecessary;

8)      “as reported by (28; 28),” - ? by who?

9)      (23; 29).” – by who?

10)   It should be Table 2 not Table 2A – the same for 2B and C – the tables should have consecutive numbers 3, 4 etc.;  and there is no information on statistical significance of the data – Tables 2A, 2B, 2C, 3 and statistics should be commented together with data they refer to, so 3.4. section should be removed;

11)   as the estimation of AA concentration according to the method of extraction was one of the main amins of the study, this aspect should be commented in Conclusions.

English grammar and style require improvements.

Author Response

Reviewer 1: 

1)      English should be improved e.g.: “contained higher contents”.

Response

According to referee’s suggestion, English has been improved through the manuscript (lines 71-100; 102-129; 322-359; 395; 522-535; 551-571).

2)      Abstract: too general, no specific results are given.

Response

According to referee’s suggestion, the suggested change was made (lines 28-32). Some results have been added to the abstract.

3)      All Latin names of plant species and genera should be written Italic font.

Response 

According to referee’s suggestion, the suggested change was made (line 38), and all Latin names of plant species and genera has been written in Italic font.

4)      The references should be formatted according to the requirements of the Journal.

Response

According to referee’s suggestion, the references has been formatted according to the requirements of the Journal. We thank the reviewer for bringing the point to our attention.

5)      Figure 1: the legend should be below the chart, and the text should be improved to be as informative as possible, it is not a scientific information that: “the baristas have chosen a specific particle size of the coffee 268 powder to obtain the desired result for each specific type of method …”.

Response

According to referee’s suggestion, the legend that connects the different colors with diverse extraction methods, can be found below the Figure 1.

In addiction the text of the Figure has been increased (Figure 1). We thank the reviewer for bringing the point to our attention.

6)      2.9.1. – why “mg Trolox equivalent (TE)/kg coffee beverage” –it should be expressed as mg TE/g dry weight of coffee beverage.

Response

According to referee’s suggestion, the sentence has been changed (section 2.9.1 and through the manuscript). Indeed, the sentence has been changed in mg Trolox equivalent (TE)/L of coffee beverage (not g dry weight) as the analysis has been performed on the coffee beverage.

7) “The results of chromatographic analysis (HPLC-MS/MS) on AA contents in the coffee samples are listed in Table 2A and presented in the form of histograms in Figure 1SA. The results in Table 2A are expressed as mean ± standard deviation of acrylamide (ng/mL).” – this is unnecessary.

Response

According to referee’s suggestion, results have been reported with the standard deviation (SD) to give more value to the analyses performed. Indeed, SD represents a suitable statistic measure that enhances the transparency, rigor, and interpretability of scientific research; it can help readers to understand the variability in data and to ensure the reliability of findings. The authors believe there is no need to remove them. We thank the reviewer for bringing the point to our attention.

8) “as reported by (28; 28),” -? by who?

Response

According to referee’s suggestion, the suggested change was made, (as reported by Strocchi et al., 2022 and by Soares et al., 2015). We thank the reviewer for bringing the point to our attention.

9) “(23; 29).” – by who?

Response

According to referee’s suggestion, the suggested change was made, (as reported by Schouten et al., 2021 and by Bagdonite et al., 2008). We thank the reviewer for bringing the point to our attention.

10)   It should be Table 2 not Table 2A – the same for 2B and C – the tables should have consecutive numbers 3, 4 etc.; and there is no information on statistical significance of the data – Tables 2A, 2B, 2C, 3 and statistics should be commented together with data they refer to, so 3.4. section should be removed.

Response

According to referee’s suggestion, the suggested change was made, Tables 2A, 2B, 2C and 3 were now converted into Tables 2, 3, 4 and 5.

In addition, the authors thank the reviewer for his/her comment. Table 2A, 2B and 2C does not report any statistical analysis result because, as stated in section 3.4, the analytes values were analyzed through the PLS-DA statistical tool. As the focus of our work was the discrimination of extraction methods based on the different analytes, we thought that applying these statistics rather than ANOVA test, which is not specific for discriminant analysis, was more suitable for our purpose.

11)   as the estimation of AA concentration according to the method of extraction was one of the main amins of the study, this aspect should be commented in Conclusions.

Response

According to referee’s suggestion, the requested information has been inserted (lines 556-569).

Reviewer 2 Report

The authors present a nice study on brewing methods and how it impacts composition of coffee.  I have several suggestions for improving the manuscript:

1. The introduction is one long paragraph.  It should be broken into more readable paragraphs.

2. The authors indicate the data in Table1 is generated from replicates, yet no statistical data is shown.  Minimally standard deviations of the averages presented in the table should be included in the table data.  The other tables in the manuscript contain the statistics, so it seems especially strange. 

3.  Figure 1 is very nice but hard to see.  I would suggest (A) "zooming in" on the relevant regions for sizes (>10 uM and above) with the full range figure in the supplement, and (B) consider splitting the figure into two or three panels to help thin out the number of overlapping curves.  

4. It is unclear why the authors chose to name the tables 2a, 2b, and 2C rather than 2-4.  If there is some reason, fine, but it just seems odd.

5. Since the units differ in each column in Table 3 (mg/kg, mg/g), i would suggest incorporating those units in the table column titles as well

6. Figure 2b and Figure 3 are so rich with data but very difficult to see.  I would suggest making these panels larger to enhance legibility.

7. The authors mention the mineral salt balance /content of the water used in the methods.  Is the salt composition of this water known?  

Author Response

Reviewer 2: 

The authors present a nice study on brewing methods and how it impacts composition of coffee. 

Response

The authors thank the reviewer for his/her comment.

I have several suggestions for improving the manuscript:

  1. The introduction is one long paragraph.  It should be broken into more readable paragraphs.

Response

According to referee’s suggestion, two readable paragraphs have been inserted (lines 43 and 70).

  1. The authors indicate the data in Table 1 is generated from replicates, yet no statistical data is shown.  Minimally standard deviations of the averages presented in the table should be included in the table data.  The other tables in the manuscript contain the statistics, so it seems especially strange. 

Response 

According to referee’s suggestion, standard deviations in Table 1 has been included yet.

For coffee powder, water temperature, time, and brew ratio, deviations have not been reported, since the coffee powder, water temperature and time always has remained invariable, same trend for the brew ratio who didn’t change between replicates.

  1. Figure 1 is very nice but hard to see.  I would suggest (A) "zooming in" on the relevant regions for sizes (>10 uM and above) with the full range figure in the supplement, and (B) consider splitting the figure into two or three panels to help thin out the number of overlapping curves.  

Response

According to referee’s suggestion, the suggested change was made, and the Figure 1 has been “zooming in”. We thank the reviewer for bringing the point to our attention.

  1. It is unclear why the authors chose to name the tables 2a, 2b, and 2C rather than 2-4.  If there is some reason, fine, but it just seems odd.

Response

According to referee’s suggestion, the suggested change has been made, Tables 2A, 2B, 2C and 3 has been now converted into Tables 2, 3, 4 and 5. We thank the reviewer for bringing the point to our attention.

  1. Since the units differ in each column in Table 3 (mg/kg, mg/g), I would suggest incorporating those units in the table column titles as well.

Response 

According to referee’s suggestion, the suggested change has been made (now Table 5).

  1. Figure 2b and Figure 3 are so rich with data but very difficult to see.  I would suggest making these panels larger to enhance legibility.

Response

According to referee’s suggestion, the quality of Figures 2b and 3 has been improved.  

  1. The authors mention the mineral salt balance /content of the water used in the methods.  Is the salt composition of this water known?  

Response

According to referee’s suggestion, the suggested information was added (lines 116-128).

Round 2

Reviewer 1 Report

The manuscript has been improved and can be accepted for publication.